# The Effects of Bariatric Surgery on Vitamin B Status and Mental Health

**DOI:** 10.3390/nu13041383

**Published:** 2021-04-20

**Authors:** Amna Al Mansoori, Hira Shakoor, Habiba I. Ali, Jack Feehan, Ayesha S. Al Dhaheri, Leila Cheikh Ismail, Marijan Bosevski, Vasso Apostolopoulos, Lily Stojanovska

**Affiliations:** 1Department of Nutrition and Health, College of Medicine and Health Sciences, United Arab Emirates University, Al Ain 15551, United Arab Emirates; 201790103@uaeu.ac.ae (A.A.M.); 201890012@uaeu.ac.ae (H.S.); habAli@uaeu.ac.ae (H.I.A.); ayesha_aldhaheri@uaeu.ac.ae (A.S.A.D.); 2Institute for Health and Sport, Victoria University, Melbourne, VIC 8001, Australia; jack.feehan@vu.edu.au (J.F.); vasso.apostolopoulos@vu.edu.au (V.A.); 3Department of Medicine-Western Health, The University of Melbourne, Melbourne, VIC 8001, Australia; 4Clinical Nutrition and Dietetics Department, College of Health Sciences, University of Sharjah, Sharjah 27272, United Arab Emirates; lcheikhismail@sharjah.ac.ae; 5Nuffield Department of Women’s & Reproductive Health, University of Oxford, Oxford OX1 2JD, UK; 6Faculty of Medicine Skopje, University Clinic of Cardiology, University of Ss. Cyril and Methodius, 1010 Skopje, North Macedonia; marijanbosevski@yahoo.com

**Keywords:** vitamin B, serotonin, dopamine, homocysteine, bariatric surgery, pro-inflammatory cytokines

## Abstract

Diet is a modifiable factor that ensures optimal growth, biochemical performance, improved mood and mental functioning. Lack of nutrients, notably vitamin B, has an impact on human health and wellbeing. The United Arab Emirates is facing a serious problem of micronutrient deficiencies because of the growing trend for bariatric surgery, including Roux-en-Y gastric bypass and sleeve gastrectomy. People undergoing bariatric surgery are at high risk of developing neurological, cognitive, and mental disabilities and cardiovascular disease due to deficiency in vitamin B. Vitamin B is involved in neurotransmitter synthesis, including γ-aminobutyric acid, serotonin, dopamine, and noradrenaline. Deficiency of vitamin B increases the risk of depression, anxiety, dementia and Alzheimer’s disease. In addition, vitamin B deficiency can disrupt the methylation of homocysteine, leading to hyperhomocysteinemia. Elevated homocysteine levels are detrimental to human health. Vitamin B deficiency also suppresses immune function, increases the production of pro-inflammatory cytokines and upregulates NF-κB. Considering the important functions of vitamin B and the severe consequences associated with its deficiency following bariatric surgery, proper dietary intervention and administration of adequate supplements should be considered to prevent negative clinical outcomes.

## 1. Introduction

Over the past three decades, the world has undergone substantial demographic, economic, political, socio-cultural, and environmental changes that are affecting diet, nutrition, and health more broadly. Due to these nutritional transitions, undernutrition coexisting with overnutrition is widely prevalent in many parts of the world [1], with estimates that micronutrient deficiencies affect more than 2 billion people globally [2]. Additionally, there is an increase in worldwide obesity due to changes in eating pattern and lifestyle. To prevent and manage obesity, bariatric surgery is often recommended to sustain weight loss [3]. Whilst helpful, bariatric surgery has some side effects, including decreased absorption of various essential nutrients such as B complex vitamins, vitamins A, D, K, iron, selenium, zinc, and copper [4,5]. There are three major types of bariatric surgery: (1) laparoscopic sleeve gastrectomy (LSG); (2) laparoscopic adjustable gastric banding; and (3) Roux-en-Y gastric bypass (RYGB). RYGB alters the gastrointestinal tract, bypassing the duodenum and jejunum, reducing nutrient absorption and metabolism [5,6]. Given that the duodenum, jejunum, and ileum are involved in vitamin B absorption, bariatric surgery could induce intestinal malabsorption of the vitamin B complex (Table 1) [4,7]. Vitamin B complex is a group of eight related vitamin Bs (vitamin B_1,2,3,5,6,7,9,12_), and deficiency in any of these is associated with a wide range of disorders. The B complex vitamins are integral to the synthesis of neurotransmitters and proper functioning of the central nervous system, and play a key role in the methylation and decarboxylation reactions necessary for the integrity and synthesis of DNA, proteins, phospholipids, monoamine, and catecholamine neurotransmitters [8,9,10]. Further, vitamin B regulates immune response by decreasing the production of pro-inflammatory cytokines, NF-κB, and nerve growth factor [11]. In addition, B vitamins, particularly folic acid (B_9_), pyridoxine (B_6_), and cobalamin (B_12_), are involved in the re-methylation and metabolism of homocysteine. High homocysteine levels contribute to neurodegenerative disorders, psychiatric disorders, and cardiovascular disease [12]. It has been shown that vitamin B deficiency leads to declines in cognitive function and causes several other mental disorders such as depression, anxiety, dementia, and Alzheimer’s disease (Figure 1) [9,11,13,14,15,16].

This narrative review aims to explore the nutritional deficiencies of vitamin B following bariatric surgery and its clinical outcomes, such as mental and cognitive problems. This paper also identifies critical strategies for managing and preventing vitamin B deficiency in bariatric surgery patients.

## 2. Methodology

Literature searches were conducted in ‘PubMed’ and ‘Google Scholar’ databases. Search terms included ‘bariatric surgery’ OR ‘gastric banding’ OR ‘laparoscopic sleeve gastrectomy’ OR ‘Roux-en-Y gastric bypass’ AND ‘micronutrients deficiency’ OR ‘vitamin B deficiency’ OR ‘vitamin B complex’ OR ‘vitamin B_1_’ OR ‘thiamine’ OR ‘vitamin B_2_’ OR ‘riboflavin’ OR ‘vitamin B_3_’ OR ‘niacin’ OR ‘vitamin B_5_’ OR ‘pantothenic acid’ OR ‘vitamin B_6_’ OR ‘pyridoxine’ OR ‘vitamin B_9_’ OR ‘folic acid’ OR ‘folate’ OR ‘vitamin B_12_’ OR ‘cobalamin’ AND ‘psychological disorder’ OR ‘depression’ OR ‘anxiety’ OR ‘bipolar’ AND ‘cognitive disorder’ OR ‘Alzheimer’ OR ‘dementia’ AND ‘neurological disorders’ OR ‘Wernicke encephalopathy’ OR ‘peripheral neuropathy’, OR ‘hyperhomocysteinemia’, with filters identifying studies published between 2000 to 2021. Irrelevant studies were excluded after examination of the title and the abstract. A total of 133 relevant studies, mainly clinical trials on bariatric surgery, have been included.

## 3. Role of Vitamin B in Human Health and the Immune System

Vitamin B_1_ (thiamine) acts as a coenzyme in the pentose phosphate pathway, which is essential for the production of fatty acids, steroids, nucleic acids, and aromatic amino acid precursors, neurotransmitters, and other bioactive compounds that are necessary for brain function [17]. Vitamin B_1_ deficiency causes overexpression of pro-inflammatory cytokines such as interleukin (IL)-1, IL-6, and tumor necrosis factor-alpha (TNF-α) as well as increased expression of CD40 and CD40 ligand by microglial cells and astrocytes, which eventually leads to the death of neuron cells [18,19]. Vitamin B_2_ (riboflavin) is derived from two flavoprotein coenzymes: flavin adenine mononucleotide and flavin adenine dinucleotide, which are important rate-limiting factors in cellular enzymatic processes [20]. Interestingly, the flavoproteins (a derivative of riboflavin) are known cofactors in the metabolism of essential fatty acids of brain lipids [21], as well as being a neuroactive compound with immunomodulatory effects. Additionally, B_2_ deficiency leads to pro-inflammatory patterns of gene expression [22] and leads to negative consequences for brain function. Further, Vitamin B_3_ (niacin) is derived from nucleotides such as nicotinamide adenine dinucleotide and nicotinamide adenine dinucleotide phosphate, which are involved in a number of body processes and enzymatic reactions [21]. Dietary niacin is primarily absorbed in the small intestine; however, a small amount can also be absorbed in the stomach [23]. Niacin is involved in the DNA metabolism and repair, cellular signaling events, cell migration [24], and decreases the expression of the pro-inflammatory cytokines, IL-1, IL-6, and TNF-α by the macrophages [25]. Low concentrations of niacin impair nicotinamide adenine dinucleotide-dependent nuclear and mitochondrial functioning, resulting in age-associated neurological disorders [26,27]. Vitamin B_5_ (pantothenic acid) is incorporated into coenzyme A that is central to a number of vital metabolic processes. Coenzyme A is needed for acetylation, an important part of a number of physiological chemical reactions, most notably in the metabolism of energy [28].

Vitamin B_6_ (pyridoxine) is a cofactor involved in carbohydrate, fat, and amino acid metabolism [29]. The phosphorylated form of B_6_ (pyridoxal phosphate) is first hydrolyzed, then absorbed and transported through carrier-mediated sodium dependent transporters [30]. B_6_ deficiency influences both the innate and adaptive immune systems, reducing the number, activity and proliferation of immune cells, impairing the growth and maturation of lymphocytes, affecting the production of antibodies by B cells, and reducing the size of the thymus gland [22]. Vitamin B_9_ (folate) is involved in the biosynthesis of nucleic acids, protein, blood cells, and the nervous system tissues [31], while B_12_ (cobalamin) is involved in DNA synthesis as well as fatty acid and amino acid metabolism [32]. Dietary vitamin B_12_ is bound to protein in food, and its absorption follows after stomach acid hydrolysis. It is then bound by the gastric R binder protein secreted in both saliva and gastric juices and passes into the duodenum. In the small intestine, detached from the R binder protein, the free B_12_ binds to intrinsic factor from the stomach’s parietal cells, allowing for absorption in the terminal ileum [33]. Folate and B_12_ deficiencies can cause T cell proliferation and influence the production of pro-inflammatory cytokines [34]. B_12_ enhances TNF-α and nerve growth factor secretion and reduces the levels of epidermal growth factor and IL-6 [35]. High levels of TNF-α and nerve growth factor can damage myelin and reduce epidermal growth factor, thus decreasing their myelinotrophic effects [36,37]. B_12_ deficiency can also adversely affect the methylation reaction, increasing inflammatory responses [38], and reducing CD8+ and natural killer cell activity [39]. The functions of vitamin B and its deficiency related outcomes are indicated in Table 1.

## 4. Bariatric Surgery and Vitamin B Deficiency

### 4.1. Bariatric Surgery

Nutritional interventions, medication, and exercise have limited effectiveness in weight loss. Therefore, obese people with body mass index (BMI) ≥ 40 kg/m^2^ have been advised to undergo bariatric surgery [48]. Bariatric surgery is a metabolic surgery associated with long-term weight loss and remission of weight-related comorbidities [49]. Some 634,897 surgical bariatric/metabolic interventions were performed worldwide in 2016 [50]. Laparoscopic sleeve gastrectomy (LSG), Roux-en-Y gastric bypass (RYGB), and gastric banding dominate the field [51]. Sleeve gastrectomy is a procedure in which 70–85% of the stomach is removed, resulting in a reduction in gastric reservoir size [51,52] and accelerated nutrient transit time, thus decreasing the absorption of nutrients [52]. In RYGB, a 30-milliliter pouch is created from the proximal stomach. The jejunum is divided, with one part attached to the artificially created pouch and the other to the duodenum to allow the pancreatic and biliary secretions to enter the intestine. The changes in the gut hormone affect eating behavior and appetite [53]. Gastric bypass affects hormones that control the body weight and eating behavior, such as ghrelin and glucagon-like peptide, while sleeve gastrectomy affects ghrelin hormone [53]. In gastric banding, a band is placed around the proximal stomach to create a small pouch to minimize the food intake without affecting the absorption [5]. Although these approaches have superior long-term weight loss results, patients are at high risk of vitamin B malabsorption following bariatric surgery (Table 2). Nevertheless, vitamin B deficiency could also exist in pre-operative stages of obesity. Therefore, vitamin B supplementation is crucial to prevent the deficiency in both pre- and post-operative stages. However, the composition of multivitamins is extremely variable. For example, vitamin B_12_ contained in multivitamin can vary from half of the recommended daily allocation (RDA) (1.2 µg/day) to 24 µg/day (10× the RDA). Furthermore, some multivitamins are designed specifically for bariatric surgery (with 250 µg–350 µg B_12_ per tablet, B_1_ (4.2 mg), B_2_ (4.8 mg), B_6_ (6 mg) [54]. Gasteyger et al. observed patients after two years of bariatric surgery, who were systematically taking multivitamins containing 2.4 µg of vitamin B_12_, that 80% of their patients had a deficit and had to be supplemented [54]. In the longer term, in a series of 75 patients followed for 83.4 ± 14.3 months (7 years) and not supplemented, 61.8% of patients had a low vitamin B_12_ level [54,55]. In another study, at 5 years, it was noted there was vitamin B_12_ deficiency in 70% of patients [55]. After restrictive surgery (gastric band), vitamin B_12_ deficiency is not uncommon and can affect 10% of patients [56], but is less harmful when patients take multivitamins. A case of vitamin B_12_ deficiency with neurological complications has been reported after gastroplasty [57].

In a comparative study between RYGB and sleeve gastrectomy (SG), the risk of vitamin B_12_ deficiency was 3.55 times higher after RYGB than after SG (95% CI, 1.26–10.01; *p* < 0.001) however, this difference disappeared when GB patients followed routine supplementation [56]. Six studies with a small number of subjects (between 9 and 60) evaluated the risk of vitamin B_12_ deficiency in between 0 and 19.6% of deficient patients, after a maximum follow-up of 36 months (reviewed in [58]). To date, no study with a follow-up of more than three years, with reliable data and a sufficient number of patients, is available to assess this long-term risk [59].

**Table 2 nutrients-13-01383-t002:** Percentage of vitamin B deficiency in bariatric surgery.

Number of Participants	Duration and Stage	Percentage (%) of Vitamin B Deficiency
232 bariatric surgery participants [60]	Post-operative	Folate (3.4%), B_12_ (18.1%), B_3_ (5.6%), B_6_ (2.2%)
169 RYGB patients [61]	Pre-operative, 1,2,3, years’ post-operative	Pre-operative B_12_ deficient (12.3%), Postoperative B_12_ after 1, 2, 3 years (19%, 28%, 29%)
149 bariatric surgery participants [62]	Post-operative	B_12_ (11%)
30 patients underwent laparoscopic RYGB [63]	6-months preoperative and 3-year post-operative	B_12_ at 2 years (33.3%) and 3 years (27.2%). No folic acid deficiency
98 participants underwent RYGB and LSG [64]	1-year pre-operative and 1-year post-operative	B_12_ deficient one-year post-operative elevated from 6.4–25.5% in the RYGB group
468 patients underwent RYGB and LSG [65]	Pre-operative and post-operative and after one year	Pre-operative B_1_ deficiency in LSG (8.1%) and RYGB (1.7%) Post-operative B_1_ deficient in LSG (10.5%) and RYGB (13.7%). One-year B_1_ deficient in LSG (7.2%) and in RYGB (5.9%).
95 participants underwent RYGB and SG [66]	Post-operative	Low level of vitamin B_12_ in RYGB (42.1%) and LSG (5%). Folate deficiency in RYGB (20%) and LSG (18.4%).
74 Gastric bypass participants [67]	>1 year	Folate (38%)
253 RYGB and 142 SG participants [68]	1–2 years post-operative	The serum concentration of vitamin B_12_ was significantly higher in the group who had undergone SG as compared to RYBG at 2 years
37 patients with severe obesity undergoing bariatric surgery [69]	3 months and 1 year post-operatively	During the year following operation, vitamin B_6_ level enhanced
60 bariatric surgery patients (gastric bypass, duodenal switch) All patients received multivitamin, and gastric bypass patients received B_12_ substitute [70]	6 months pre-operative, and 1 year post-operative	Duodenal switch patients showed thiamine deficiency after surgery. The level of riboflavin and vitamin B_6_ did not change after surgery
1160 subject with RYGB, 883 received, and 258 did not receive, specialized multivitamin supplements [71]	3 years post-operative	Participants who received specialized multivitamin supplements were less deficient in vitamin B_12_, vitamin D, folic acid, and ferritin as compared to other group receiving no supplements
45 Bariatric patients treated with intramuscular hydroxocobalamin injections, while 45 did not receive [72]	Post-operative	The treated group reported significantly increased vitamin B_12_ and showed fewer clinical complaints
1538 patients’ micronutrient status assessed prior to bariatric surgery [73]	Pre-operative	Vitamin B_12_ deficiency was 16%, and various other micronutrient deficiencies pre-exit High level of vitamin B_6_ by 24% found before surgery
103 morbidly obese women before bariatric surgery [74]	Pre-operative	10.6% of participants had B_12_ deficiency, No folic acid deficiency Deficiency of other micronutrients (iron, zinc, calcium, phosphorus)
1732 patients with morbid obesity wishing to undergo bariatric surgery [75]	Pre-operative	63.2% of participants had a folic acid deficiency and various other micronutrient deficiencies
2008 morbid obese participants wanted bariatric surgery [76]	Pre-operative	Participants deficient in vitamin D, vitamin B_12_, iron, and hemoglobin by 53.6%, 34.4%, 10.2%, and 16.6%, respectively, prior to bariatric surgery
114 patients assigned for bariatric surgery [77]	Pre-operative	Participants deficient in iron, folic acid, ferritin, vitamin B_12,_ and calcium by 35%, 24%, 24%, 3.6%, and 0.9%, respectively, prior to bariatric surgery
200 patients with SG [78]	Pre- and Post-operative	Participants deficient in B_1_, B_6_, B_12_, folic acid, vitamin D by 5.5%, 3%, 11.5%, 24%, and 81, respectively, prior to surgery and deficient after surgery in B_1_, B_6_, B_12,_ and vitamin D by 9%, 4%, 11.5%, and 36%, respectively

Abbreviations: LSG, Laparoscopic sleeve gastrectomy; RYGB, Roux-en-Y gastric bypass; SG, sleeve gastrectomy.

### 4.2. Mechanisms of Vitamin B Deficiency Following Bariatric Surgery

The degree to which bariatric surgery can cause vitamin B deficiencies depends mainly on the particular type of operation performed [52,53]. Therefore, assessing bariatric surgery patients’ nutritional consequences should be based on the surgical procedure type [53].

The gastrointestinal tract’s physiological and anatomical changes accompanying gastric bypass mainly result in vitamin B_9_ and B_12_ malabsorption [79]. Therefore, vitamin B_12_ deficiency is more commonly associated with the RYGB procedure [51,64,80], however, it is present after both procedures [81]. Evaluation of the long-term impact of RYGB on B_12_ status showed an increase in B_12_ deficiency from 2.3% at the baseline to 6.5% at 12 months following the surgery [82]. Likewise, B_12_ was significantly lower in the RYGB group compared to the LSG group [82,83]. B_12_ deficiency onsets rapidly, with changes in absorption present as little as two months after the surgery, with associated increases in homocysteine levels [81]. Absorption of B_12_ depends on the intrinsic factor, which is almost in the gastric bypass population [84]. 35% of RYGB patients in the Lakhani study experienced bacterial growth syndrome, an important factor that can lead to B_12_ deficiency [83]. Notably, approximately 12% of bariatric surgery candidates are already B_12_ deficient before their operation, likely worsening deficiency following the operation [61]. Oher factors that lead to B_12_ deficiency are intolerance of dietary meat intake, which is the primary source of vitamin B_12_, and a reduction of intrinsic factor in the terminal ileum, which is essential for B_12_ absorption [79]. A systematic review assessing the relationship between bariatric surgery and diet quality noted that those receiving gastric banding were more likely to experience gastrointestinal symptoms and food intolerances than sleeve gastrectomy and RYGB populations during the first year. Besides, the SG population had a better food intolerance than RYGB [85]. Additionally, the alteration in the stomach acid and pepsin enzyme secretion that accompanies gastric bypass interferes with cobalamin absorption [86].

Some gastric banding patients may experience recurrent vomiting due to the banding [62], and prolonged and aggressive vomiting occurring following bariatric surgery can lead to thiamine deficiency in this population [87,88,89]. Further, 35% to 65% of patients experience hyperemesis following the operation due to feelings of fullness or digestive tract plugging, which exacerbates B_1_ deficiency in bariatric surgery patients [53]. The sleeve gastrectomy patients are more likely to experience thiamin deficiency [51,64,80]. As opposed to the narrow zone of absorption of B_12_, folate absorption occurs along the whole length of the small intestine [80]; therefore, folate deficiency in this population was primarily attributed to non-adherence to supplementation rather than malabsorption. Patients who adhered to an 800 µg of folic acid daily did not experience folate deficiency [5]. Indeed, RYGB patients are more likely to experience vitamin B_12,_ while SG and gastric banding patients are more likely to experience thiamine deficiency.

Non adherence to supplements contributes to worsening the micronutrient deficiency among bariatric surgery patients [90]. The deficiency of micronutrients is higher in non-adherence bariatric surgery patients than adherence bariatric surgery patients [91,92]. Bariatric surgery-related factors include vomiting, which is a common postoperative complication seen in 30% of SG patients [53], and the disturbed eating habit leads to difficulty in supplement adherence. Likewise, difficulty swallowing drugs and forgetting to take supplements are barriers to supplement adherence among bariatric surgery patients [93]. The high cost of particular multivitamins for bariatric surgery patients contributes to long-term non-adherence to the supplement [94].

### 4.3. Bariatric Surgery and Hyperhomocysteinemia

Bariatric patients with vitamin B deficiency are at risk of secondary neuropsychological disorders and CVD [91]. The malabsorption of vitamin B _9_ and B_12_ following bariatric surgery affects the re-methylation pathway of homocysteine, leading to hyper-homocysteinemia [83]. Increased levels of homocysteine beyond 15 µmol/L are expected among bariatric surgery patients [86]. Elevated homocysteine level has been detected in 29% of bariatric surgery patients [62]. A high homocysteine level is found in the bypass group (10.4 µmol/L), compared to (9.2 µmol/L) in control [95]. Likewise, a study showed a higher homocysteine level in patients after the surgery (14.6 µmol/L) compared to (11.6 µmol/L) at the baseline values before the operation [79]. Multivitamin supplements lower the homocysteine level [95]. If methylation reactions are limited, it can lead to a range of problems such as anxiety, depression, bipolar disorder, Alzheimer’s disease (AD), schizophrenia, and sleep-cycle disturbance [96,97] (Figure 2).

### 4.4. Depression and Anxiety in Patients Following Bariatric Surgery

Vitamins B has an essential role in synthesizing neurotransmitters and factors such as serotonin that affect mood and other brain functions. Vitamin B deficiencies, particularly B_1_, B_6_, B_9,_ and B_12_, are known causes of psychiatric disorders, including depression and dementia [98]. Patients undergoing bariatric surgery commonly experience thiamin deficiency, crucial for Thiamin-dependent enzyme function [99]. Thiamin-dependent enzymes play a critical role in glucose metabolism, which is essential to ensure optimal brain function. Further, the brain is susceptible to thiamin availability, and it has been shown that glucose levels are diminished in 20–30% of brain regions in those living with Alzheimer’s [99].

Clinically, mental issues are common in severely obese adults seeking bariatric surgery [100,101,102,103,104,105], and depression is the most crucial issue in this population [102]. Depression exists in 45.2% of bariatric surgery candidates and 2.7% show severe depression [101]. In this study, the Beck score improved at six months compared with the baseline score, and the improvement continues during the first year. A longitudinal assessment of bariatric surgery, including 2148 patients, showed an improvement in depression symptoms during the first year following bariatric surgery; then it deteriorated during 1–3 years following bariatric surgery [100].

Furthermore, bariatric surgery procedure affects the hospitalization rate for depression. When assessing this comparing RYGB and LAGB, 1.2% of RYGB reported hospitalization for depression compared with 0% of LAGB. Further, at three years from surgery, the rate of RYGB admitted for depression increased to 2.1% compared with 0.6%. They attributed this deterioration of mental health to the BMI change [100]. Further, the severity of depression symptoms before surgery predicts the postoperative BDI score. A longitudinal assessment of bariatric surgery study revealed patients with moderate, severe depression symptoms at baseline had 7.8 higher odds of having a moderate-severe symptom of depression. In contrast, those with minimal symptoms had 6.77 higher odds of experiencing mild depression symptoms. 35.3% of bariatric surgery candidates reported taking at least one antidepressant medication, and serotonin reuptake inhibitors (SSRI) are the most common antidepressant [100]. Serotonin reuptake inhibitor level dropped in RYGB patients after one month, and 54% of patients relapsed after one month of the surgery [106]. The diminished intestinal surface that reduces the drug’s exposure to the absorption area as in RYGB negatively affects bariatric surgery patients’ drug disposition [106].

Likewise, after bariatric surgery, psychological outcomes showed an improvement in depression symptoms at 6, 12, 24 months post-operatively, and after 24 months, it returned to the baseline levels [107]. Further, bariatric surgery patients scored higher in healthy quality of life post-operatively and scored lower after the weight-stability phase [53]. In a case report, a 38-year-old male patient who had undergone bariatric surgery developed episodes of psychotic depression, which was attributed to vitamin B_12_ deficiency. Supplementing vitamin B_12_ caused remission of the patient’s clinical symptoms of depression [108]. Conversely, one hundred RYGB participants using the Beck Depression inventory reported a worsening mood in 3.7% of participants during the 6 to 12 months following the operation [109]. Given the well-known physiological consequences of vitamin B deficiency and its common presentation in those who have undergone bariatric surgery, it is a plausible mechanism underlying the onset of depression and anxiety in these patients.

### 4.5. Neurological Complications in Patients Following Bariatric Surgery

About, 4.6–16% of bariatric surgery patients experience postoperative neurological complications [87]. Thiamin deficiency is common following bariatric surgery [110] due to the aggressive vomiting attacks in this population [87], commonly seen in so called ‘bariatric beriberi’ [89]. Thiamine deficiency commonly manifests as neurological complications, including Wernicke encephalopathy (WE) [88,89]. A study of 100 cases of Wernicke Encephalopathy (WE) following bariatric surgery found that onset of the WE symptoms varied according the surgery type, and RYGB is associated more with neurological complication [89]. The most consistent factor among WE patients was persistent vomiting [89,111]. Bariatric surgery with high BMI seems to have less severe WE symptoms, explained by “preferential intracellular thiamin cycling”, suggesting rapid weight loss and depleted thiamin store, resulting in WE symptoms [89].

Peripheral neuropathy is a common neurological complication in this population [88,112]. The rapid weight loss accompanying bariatric surgery leads to neuropathy compression by exposing the nerve to subcutaneous tissue loss [87,113] Intravenous thiamin replacement therapy contributes to reverse neurological symptoms at early stages. 93.3% of bariatric surgery patients recovered fully from neurological complications [87]. The neurological disorder was reversed if they were diagnosed early [4]. A study assessing the neurological clinical manifestation noted that B_1_, B_2_, B_6_ and B_12_ deficiencies were common amongst RYGB and SG who showed neurological clinical manifestations including paresthesia muscle weakness and abnormal gait [4].

## 5. Effectiveness of Vitamin B Supplementation in Bariatric Surgery Patients

Vitamin B supplements are a key means of meeting the body’s vitamin B needs in individuals following bariatric surgery [4,83]. Currently, the B_12_ doses in over-the-counter multivitamin formulations are insufficient to meet post-operative patient needs [5,83,114], and as such tailored doses and administration of B supplementation are crucial considerations for these patients [5,71,83,114]. The dose of B supplementation depends predominantly on the surgery type [83]. High doses of oral cyanocobalamin are ideal for RYGB patients, while lower doses of vitamin B_12_ should be enough for those receiving LSG and gastric banding [80,83]. The recommended dose of B_12_ by the British Obesity and Metabolic Surgery Society is inadequate in RYGB due to the high prevalence of B vitamin deficiencies in this group [83]. Therefore, they require a higher dose than the RDA to meet the increased demand for vitamin B_12_. Vitamin B_12_ (350 μg/day) should be the minimum dose after gastric bypass and can be administered orally or parentally [5]. 1000 μg, and 2000 μg doses of vitamin B_12_ have been suggested for optimal absorptive capacity [71].

Oral, intramuscular, intranasal, intravenous, and enteral parental are the suggested routes to administrate vitamin B. A systematic review of randomized controlled trials assessed the efficacy of oral B_12_ and intramuscular B_12_ injection. A high serum level of B_12_ was observed in an oral group when given in high doses compared with the intramuscular group at 2 and 4 months follow up [115]. Likewise, a systematic review showed a B_12_ of dose <15 µg is insufficient to correct B_12_ serum level in RYGB patients, and 10,000 µg showed superior results and increased B_12_ serum level hence, preventing B_12_ deficiency in this cohort [114]. A meta-analysis of studies comparing oral and parenteral routes showed in 108 gastrectomized or achlorhydric patients deficient in vitamin B_12_ that the oral route was as effective, or even faster, than the intramuscular route on the condition of 1000 to 2000 µg/day of vitamin B_12_ during the first weeks, then weekly and monthly [116]. Pharmacokinetic studies show that approximately 1% of a dose >25 µg of crystalline form is absorbed passively, or 10 µg for a dose of 1000 µg [117], yet the RDA for this vitamin is 2.4 µg/j. Further, the crystalline form of B_12_ showed its efficacy for absorption in the absence of intrinsic factor when given in high doses [118]. In contrast, a study showed that an oral supplement is not enough to correct the serum level of B_12_ in gastric bypass patients [119]. The British Obesity and Metabolic society recommend intramuscular injection of vitamin B_12_ should be taken every three weeks in RYGB and BPD/DS patients, since the deficiency of B_12_ still exists in the presence of higher doses of the oral supplement [120]. The intramuscular route is used for those who have severe B_12_ deficiency symptoms, have gastrointestinal intolerance, and are not compliant with their oral supplementation, or when oral supplementation does not maintain B_12_ level. Intravenous B_12_ is associated with anaphylactic shock, while nasal and sublingual routes are under evaluation for their efficiency [83].

Following GB, the doses to be prescribed are not clearly defined. In some studies, a dose of 350 µg/day is adequate to maintain plasma levels [118]. A dose of 1000 µg/week seems sufficient, a fraction of this contribution being able to be absorbed by the intrinsic factor [116,118,121]. This dose must sometimes be doubled and recourse to the intramuscular route should be preferred only when patients are not very observant. This strategy has been validated in randomized study [122], but most of the studies recommend supplements through the oral route [123,124,125,126,127,128,129].

## 6. Strategies to Prevent Vitamin B Deficiency in Bariatric Surgery Patients

Given B deficiency is caused by the anatomical changes accompanying bariatric surgery, vitamin B status provision is imperative. Managing vitamin B status among BS patients has been divided into pre-operative, postoperative phase (<5 days), and postoperative phase (>5 day) [116].

Vitamin B deficiency could exist in the pre-operative stage; 20–30% of BS candidates have micronutrient deficiencies before surgery [88]. Early detection is important to identify vitamin B deficiency. Therefore, vitamin B supplementation is very important to prevent the deficiency in both pre- and post-operative stage. The pre-operative period is critical to emphasize the importance of vitamin adherence [83,130], and inform the patients about the side effects of vitamin B deficiency. Assessing B_12_ status in bariatric surgery patients requires a reliable test that reflects the B_12_ status. Homocysteine and methylmalonic acid (MMA) are the two reliable tests that reflect B_12_ status [131]. Since B_12_ deficiency results in MMA accumulation before B_12_ reduction is seen in the serum [132], methylmalonic acid is a better indicator and preferred marker reflecting B_12_ status before B_12_ deficiency appears [120].

Patients may experience nausea, vomiting, and dumping syndrome; therefore, administrating, and replenishing BS patients with vitamin B supplementation immediately after the operation is crucial to avoid thiamin deficiency [109]. As some deficiencies (particularly vitamin B_12_) may take years to present clinical symptoms, the long-term management of vitamin B deficiency is crucial. Constant follow-up and lifelong mineral and multivitamin supplementation are recommended [5,53,133].

## 7. Conclusions

There is a strong relationship between nutritional deficiencies and disease. Patients who receive bariatric surgery are at high risk of developing mental, cognitive, and neurological complications resulting from micronutrient deficiencies. Recognition of the clinical presentations of vitamin B deficiency is vital, enabling early intervention and minimizing long-term adverse effects. A primary clinical concern that needs to be addressed is the relationship between vitamin B deficiency and the development of depression, anxiety, and other neurological complications. Vitamin B supplements lessen the impact of these conditions and quality of life. Further studies are needed to determine optimal vitamin B supplements in patients following bariatric surgery to minimize adverse clinical outcomes. Providing awareness regarding healthy eating habits and lifestyle changes to reduce obesity are needed. There is also a need for ongoing monitoring of these patients to avoid bariatric surgery’s unwanted side effects. Early dietary and lifestyle intervention should be implemented to reduce obesity and avoid post-operative deficiency and its associated side-effects. This will lead to a decrease in the growing prevalence of vitamin B deficiency while improving patients’ outcomes post-bariatric surgery.

## Figures and Tables

**Figure 1 nutrients-13-01383-f001:**
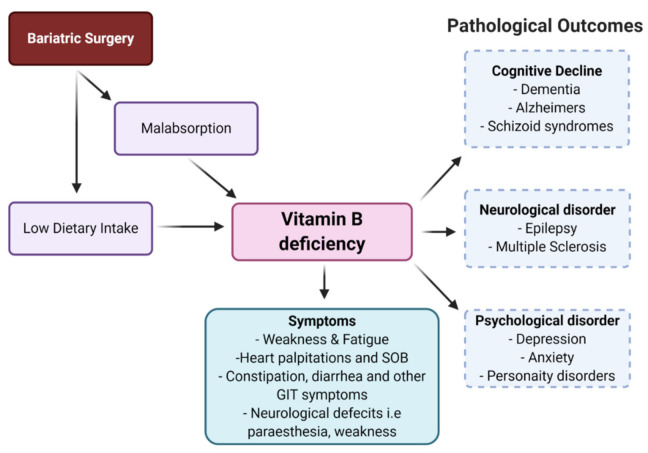
Symptoms and outcomes of vitamin B deficiency.

**Figure 2 nutrients-13-01383-f002:**
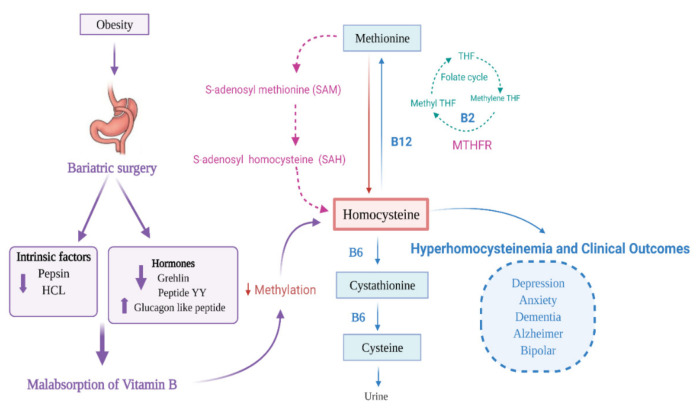
Bariatric surgery, hyper-homocysteinemia, and importance of vitamin B.

**Table 1 nutrients-13-01383-t001:** The functions of B vitamins, site of absorption, and deficiency related outcomes.

B Vitamins	Functions	Absorption Site	Deficiency Related Outcomes
B1 (Thiamine) [9,22]	Acetylcholine production, action potential generation, structure and function of cellular membranes	Duodenum Jejunum	Reduces enzymatic activity and energy production, alters mitochondrial activity
B2 (Riboflavin) [40]	Maintains the integrity of mucous membranes, skin, eyes, and the nervous system	Duodenum Jejunum	Mitochondrial dysfunction, effects one-carbon metabolism
B3 (Niacin) [41,42]	Acts as an antioxidant, produces energy, protects against axonal damage, neuroprotective role	Duodenum Jejunum	Increases oxidative stress and inflammatory cytokines, mitochondrial dysfunction
B5 (Pantothenic acid) [43]	Regulates iron by transporting oxygen to the brain, synthesizes neurotransmitters, helps in the synthesis and regeneration of myelin	Jejunum	Increased cell stress and translocation of NF-κB, altered fatty acid metabolism
B6 (Pyridoxine) [22,28]	Assists in the synthesis of hemoglobin, neurotransmitters, DNA methylation, and homocysteine metabolism	Jejunum	Altered tryptophan and one-carbon metabolism
B9 (Folate) [44]	Synthesizes norepinephrine, dopamine, and serotonin. Involved in methylation of homocysteine to methionine	Duodenum Jejunum Ileum	Disrupts DNA methylation and alters nitric-oxide balance in the blood
B12 (Cobalamin) [45,46,47]	Synthesizes new cells, involved in nerve cells maintenance, assists in breaking fatty acids and amino acids	Ileum (terminal only)	Effects on DNA synthesis, adverse effects on brain function

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
