# Peer review of "The Effects of Bariatric Surgery on Vitamin B Status and Mental Health"

_nutrients, 2021, doi:10.3390/nu13041383_

Round 1

Reviewer 1 Report

Dear authors,

Thank you for the opportunity to review your manuscript, which is a comprehensive review of vitamin B and its effects on mental disorders after bariatric surgery. I have a few suggestions;

1) Please rename the title of the manuscript. 'Effects on health' is a bit vague, while you describe something different in your introduction section (effects on mental health, e.g. disorders).

2) I'm not sure but is this a systematic review or a narrative review. If a systematic review please adhere to the PRISMA guidelines. If a narrative review please state that in the manuscript, since methodologically these two are very different.

3) Please a paragraph on the value of Methylmalonic acid in screening for vitamin B deficiencies.

4) Please add a recent systematic review and study about different vitamin B supplementation regimes after bariatric surgery. ( doi: 10.1007/s11695-016-2449-9; doi: 10.1007/s11695-016-2207-z.).

5) Also it needs to be taken into account that compliance with taking multivitamin supplements after bariatric surgery is a major factor, please a small paragraph to your manuscript (doi: 10.1017/jns.2020.41.)

Author Response

We thank the reviewer for their excellent feedback, which has been incorporated into the manuscript. Please see the attached document for a point-by-point response.

Reviewer 2 Report

Thank you for the submission. This is an interesting review which discusses an important issue related to certain bariatric surgery procedures. My comments/suggestions are below:

  1. In reference to table 2 (% of vitamin B deficiency in bariatric surgery), were confounding factors such as dietary changes post-surgery as well as existing pre- surgical deficiencies assessed and taken into account for these study findings? These factors would be important to consider as post bariatric surgery also suggests dietary intake reduction, which could lead to the patient’s nutritional intake of vitamin B decreasing, leading to a deficient, not necessarily an effect of only the decreased absorption due to the procedure itself.  Also, as you reference in section 7 (line 258), 20-30 % of patients pre-operatively are already deficient in micronutrients, which would affect the validity of the results if not taken into account.  This should be discussed briefly.
  2. In section 5.4, when discussing depression and anxiety rates post-surgery, I think it is difficult to speculate that the level of depression increased (or returned to pre-surgery levels) as a result of vitamin B deficiency related to the surgery without ruling out potential confounding factors.  As discussed, many patients pre-surgery suffer from depression.  While they tend to see a decrease in depressive symptoms acutely post-surgery, this can increase with time.  The increase or return of depressive symptoms with time could be due to other factors including post-operative results (ie are the patients losing weight as expected or were they not happy with the results? Did they end up gaining weight back with time? Both of these things could affect their mental health/ depression ).  Additionally, it would be beneficial to assess whether or not the patients have chronic depression prior to surgery, which would increase their likelihood of struggling with long-term depression, even if they have a temporary decrease of symptoms immediately after surgery.  If confounding factors for depression are not noted in the studies cited, it should be discussed as a limitation.
  3. Line 226, correct typo to *Beck Depression inventory
  4. In section 5.1, reference 48 does not appear to discuss the text it is citing. Please remove citation as reference 49 does support this statement.

Author Response

(The authors gave the same response as above.)

Reviewer 3 Report

This review of the literature is well written and of high quality. However, I have several comments.

  • Literature searches focalized on Psychological disorder’ OR ‘Depression’ OR ‘Anxiety’ OR ‘Cognitive disorder’ OR ‘Alzheimer’ OR ‘Dementia, but this review goes beyond these themes.
  • This review of the literature focuses on vitamin B12, which is justified because of the high prevalence of deficiency after bariatric surgery. However, although exceptional, vitamin B1 deficiency has sometimes dramatic neurological consequences which should also be detailed. The Gayet Wernicke encephalopathy, associated or not with a Korsakoff syndrome and polyneuropathy are reported however the prevalence of neurological form is probably largely underestimated.
  • In the same way, the implication of vit B2, B3, B5, and B6 are not sufficiently detailed.
  • Prevalence of Vitamin B deficiency analysis. The literature review must take into account the systematic supplementation prescribed to the patient and the doses present in the multivitamins. Indeed, the composition of multivitamins is extremely variable. For example, vitamin B12 contained in multivitamin can vary from 0.5 RDA (1.2µg / d) to 24µg / d (10 XRDA). Furthermore, some multivitamins are adapted specifically for bariatric surgery (with 250µg-350 µg B12/tablet [1], high dose of Vit B1, etc…). So, the prevalence presented in Table 2 is seriously underestimated the risk of deficiency in case of total absence of specific supplementation. It is therefore important to take into account the oldest studies, of populations with no supplementation or with supplementation corresponding to the level of RDA.

- 2 years after bariatric surgery, the Swiss team of Gasteyger et al. observed in patients systematically taking multivitamins containing 2.4µg of vitamin B12, that 80% of their patients had a deficit and had to be supplemented [1].

- In the longer term, in a series of 75 patients followed for 83.4 +/- 14.3 months (ie 7 years) and not supplemented, 61.8% of patients had a low vitamin B12 level [2] [1].

- At 5 years old, Halverson et al. also noted a vitamin B12 deficiency in 70% of his patients [2].

- After restrictive surgery (gastric band), vitamin B12 deficiency is not uncommon, it can affect 10% of patients [4] but is less when patients take multivitamins. A case of vitamin B12 deficiency with neurological complications has been reported after gastroplasty [3].

- In my opinion, after sleeve gastrectomy, current data are insufficient to assess the risk of deficiency. In a comparative study between CCG and sleeve gastrectomy (SG), the risk of vitamin B12 deficiency was 3.55 times higher after CCG than after SG (95% CI, 1.26-10.01; P <.001) however, this difference disappeared when GB patients followed routine supplementation [4]. Six studies with a small number of subjects (between 9 and 60) evaluated the risk of vitamin B12 deficiency, between 0 and 19.6% of deficient patients, after a maximum follow-up of 36 months (reviewed in [5]). To date, no study with a follow-up of more than three years, with reliable data and a sufficient number of patients is available to assess this long-term risk [6].

  • I do not agree with the statement "intramuscular injection of vitamin B12 is 251 considered standard therapy to prevent B12 deficiency". From the first studies on vitamin B12 absorption after GB, Rhodes et al. [8] found that vitamin B12 taken in crystalline form (cyanocobalamin) was absorbed even in the absence of intrinsic factor. Indeed, oral supplementation is possible, even in the absence of intrinsic factor, when vitamin B12 is provided in crystalline form and in high doses. A meta-analysis of studies comparing oral and parenteral routes showed in 108 patients, gastrectomized or achlorydric and deficient in vitamin B12 that the oral route was as effective, or even faster than the intramuscular route (IM) on condition of giving 1000 to 2000 µg / d of vitamin B12 during the first weeks, then weekly and monthly [9]. Pharmacokinetic studies show that approximately 1% of a dose> 25 µg of crystalline form is absorbed passively, or 10 µg for a dose of 1000 µg [7], yet the RDA for this vitamin is 2.4 µg / j.
  • After GB, the doses to be prescribed are not clearly defined. For some authors, a dose of 350 µg / day would be necessary to maintain the plasma level [8]. A dose of 1000 µg / week taken once a week seems sufficient, a fraction of this contribution being able to be absorbed by the intrinsic factor [8-10]. This dose must sometimes be doubled and recourse to the IM route should be preferred only when patients are not very observant. This strategy has been validated in randomized study (ex [11]), and Most recommendations recommend the oral route [12] [13]  [14] [15] [16] [17] [18]

Other remarks:

  • The same reference is cited twice: 53 and 66: Ziegler. 66 could be replaced by [9]

Références

[1] Gasteyger C, Suter M, Gaillard RC, Giusti V. Nutritional deficiencies after Roux-en-Y gastric bypass for morbid obesity often cannot be prevented by standard multivitamin supplementation. Am J Clin Nutr. 2008;87:1128-33.

[2] Halverson JD. Metabolic risk of obesity surgery and long-term follow-up. Am J Clin Nutr. 1992;55:602S-5S.

[3] Moschos M, Droutsas D. A man who lost weight and his sight. Lancet. 1998;351:1174.

[4] Kwon Y, Kim HJ, Lo Menzo E, Park S, Szomstein S, Rosenthal RJ. Anemia, iron and vitamin B12 deficiencies after sleeve gastrectomy compared to Roux-en-Y gastric bypass: a meta-analysis. Surg Obes Relat Dis. 2014;10:589-97.

[5] Eltweri AM, Bowrey DJ, Sutton CD, Graham L, Williams RN. An audit to determine if vitamin b12 supplementation is necessary after sleeve gastrectomy. Springerplus. 2013;2:218.

[6] Gagner M, Deitel M, Erickson AL, Crosby RD. Survey on laparoscopic sleeve gastrectomy (LSG) at the Fourth International Consensus Summit on Sleeve Gastrectomy. Obes Surg. 2013;23:2013-7.

[7] Allen LH. How common is vitamin B-12 deficiency? Am J Clin Nutr. 2009;89:693S-6S.

[8] Rhode BM, Arseneau P, Cooper BA, Katz M, Gilfix BM, MacLean LD. Vitamin B-12 deficiency after gastric surgery for obesity. Am J Clin Nutr. 1996;63:103-9.

[9] Quilliot D, Coupaye M, Ciangura C, Czernichow S, Salle A, Gaborit B, et al. Recommendations for nutritional care after bariatric surgery: Recommendations for best practice and SOFFCO-MM/AFERO/SFNCM/expert consensus. J Visc Surg. 2021;158:51-61.

[10] Nuzzo A, Czernichow S, Hertig A, Ledoux S, Poghosyan T, Quilliot D, et al. Prevention and treatment of nutritional complications after bariatric surgery. Lancet Gastroenterol Hepatol. 2021;6:238-51.

[11] Schijns W, Homan J, van der Meer L, Janssen IM, van Laarhoven CJ, Berends FJ, et al. Efficacy of oral compared with intramuscular vitamin B-12 supplementation after Roux-en-Y gastric bypass: a randomized controlled trial. Am J Clin Nutr. 2018;108:6-12.

[12] Kumari A, Nigam A. Bariatric Surgery in Women: A Boon Needs Special Care During Pregnancy. J Clin Diagn Res. 2015;9:QE01-5.

[13] Wax JR, Pinette MG, Cartin A, Blackstone J. Female reproductive issues following bariatric surgery. Obstet Gynecol Surv. 2007;62:595-604.

[14] Kaska L, Kobiela J, Abacjew-Chmylko A, Chmylko L, Wojanowska-Pindel M, Kobiela P, et al. Nutrition and pregnancy after bariatric surgery. ISRN Obes. 2013;2013:492060.

[15] Fullmer MA, Abrams SH, Hrovat K, Mooney L, Scheimann AO, Hillman JB, et al. Nutritional strategy for adolescents undergoing bariatric surgery: report of a working group of the Nutrition Committee of NASPGHAN/NACHRI. J Pediatr Gastroenterol Nutr. 2012;54:125-35.

[16] Kominiarek MA. Preparing for and managing a pregnancy after bariatric surgery. Semin Perinatol. 2011;35:356-61.

[17] Harris AA, Barger MK. Specialized care for women pregnant after bariatric surgery. J Midwifery Womens Health. 2010;55:529-39.

[18] Woodard CB. Pregnancy following bariatric surgery. J Perinat Neonatal Nurs. 2004;18:329-40.

Author Response

(The authors gave the same response as above.)

Round 2

Reviewer 1 Report

Dear authors, Thank you very much for addressing my previously given comments. 

Author Response

We thank the reviewer for their feedback.

Reviewer 3 Report

Thank you for taking my comments and corrections into account.

Author Response

(The authors gave the same response as above.)
